# Effectiveness of Telephone Monitoring in Primary Care to Detect Pneumonia and Associated Risk Factors in Patients with SARS-CoV-2

**DOI:** 10.3390/healthcare9111548

**Published:** 2021-11-13

**Authors:** Jose Miguel Baena-Díez, Isabel Gonzalez-Casafont, Sara Cordeiro-Coelho, Soledad Fernández-González, Migdalia Rodríguez-Jorge, Clara Uxía Fernández Pérez-Torres, Andrea Larrañaga-Cabrera, Manel García-Lareo, Ana de la Arada-Acebes, Esther Martín-Jiménez, Almudena Pérez-Orcero, Rosario Hernández-Ibáñez, Ana Gonzalo-Voltas, Noemí Bermúdez-Chillida, Consuelo Simón-Muela, Guillermo del Carlo, Carolina Bayona-Faro, Cristina Rey-Reñones, Isabel Aguilar-Palacio, María Grau

**Affiliations:** 1La Marina Primary Care Center, Catalan Institute of Health (ICS), 08038 Barcelona, Spain; josemibaena@gmail.com (J.M.B.-D.); igonzalezcasafont.bcn.ics@gencat.cat (I.G.-C.); sara-coelho@hotmail.com (S.C.-C.); msfernandez89@gmail.com (S.F.-G.); migda.med@gmail.com (M.R.-J.); clara_fpt@hotmail.com (C.U.F.P.-T.); alarranaga.bcn.ics@gencat.cat (A.L.-C.); 28306mgl@comb.cat (M.G.-L.); adelaarada.bcn.ics@gencat.cat (A.d.l.A.-A.); essther.martin@gmail.com (E.M.-J.); aperezor.bcn.ics@gencat.cat (A.P.-O.); rohernandez.bcn.ics@gencat.cat (R.H.-I.); anagonzalov.bcn.ics@gencat.cat (A.G.-V.); nbermudez.bcn.ics@gencat.cat (N.B.-C.); csimon.bcn.ics@gencat.cat (C.S.-M.); gfdelcarlo.bcn.ics@gencat.cat (G.d.C.); cbayona.bcn.ics@gencat.cat (C.B.-F.); 2IDIAP Jordi Gol, Catalan Institute of Health (ICS), 08007 Barcelona, Spain; crey.tgn.ics@gencat.cat; 3IMIM-Institut Hospital del Mar d’Investigacions Mèdiques, 08003 Barcelona, Spain; 4Unit of Research Support Camp de Tarragona, Catalan Institute of Health (ICS), 43202 Tarragona, Spain; 5Research Group in Health Services of Aragon, (GRISSA) IIS Aragón, University of Zaragoza, 50009 Zaragoza, Spain; iaguilar@unizar.es; 6Serra Húnter Fellow, Department of Medicine, University of Barcelona, 08036 Barcelona, Spain; 7Biomedical Research Consortium in Epidemiology and Public Health (CIBERESP), 08003 Barcelona, Spain

**Keywords:** SARS-CoV-2, COVID-19, pneumonia, epidemiology, telemedicine, telehealth, primary health care

## Abstract

Improved technology facilitates the acceptance of telemedicine. The aim was to analyze the effectiveness of telephone follow-up to detect severe SARS-CoV-2 cases that progressed to pneumonia. A prospective cohort study with 2-week telephone follow-up was carried out March 1 to May 4, 2020, in a primary healthcare center in Barcelona. Individuals aged ≥15 years with symptoms of SARS-CoV-2 were included. Outpatients with non-severe disease were called on days 2, 4, 7, 10 and 14 after diagnosis; patients with risk factors for pneumonia received daily calls through day 5 and then the regularly scheduled calls. Patients hospitalized due to pneumonia received calls on days 1, 3, 7 and 14 post-discharge. Of the 453 included patients, 435 (96%) were first attended to at a primary healthcare center. The 14-day follow-up was completed in 430 patients (99%), with 1798 calls performed. Of the 99 cases of pneumonia detected (incidence rate 20.8%), one-third appeared 7 to 10 days after onset of SARS-CoV-2 symptoms. Ten deaths due to pneumonia were recorded. Telephone follow-up by a primary healthcare center was effective to detect SARS-CoV-2 pneumonias and to monitor related complications. Thus, telephone appointments between a patient and their health care practitioner benefit both health outcomes and convenience.

## 1. Introduction

The severe acute respiratory syndrome coronavirus 2 (SARS-CoV-2) outbreak in China’s Hubei province in 2019 quickly spread throughout the world, with staggering medical, social and economic consequences [1]. While millions of cases have been confirmed worldwide, Spain has been one of the most affected countries [2], particularly in the most socioeconomically deprived communities [3]. Due to the magnitude of the pandemic and the lack of means for fast diagnosis at primary health care settings during the first wave (March–April 2020), the primary care strategy was mainly based on three pillars [4]. First, identification of non-severe cases of SARS-CoV-2 without alarming symptoms (e.g., dyspnea, fever and diarrhea) [5] that could receive at-home treatment of symptoms with quarantine and isolation. Second, identification of severe cases with alarming symptoms that should be attended in-hospital. Third, post-discharge follow-up of hospitalized patients to identify potential complications. To strengthen lockdown precautions and avoid the exposure of other patients and healthcare professionals, a telemedicine strategy (i.e., telephone follow-up) was prioritized. A growing body of evidence supports the safety and efficacy of telemedicine, showing equivalencies to a conventional medical appointment both in diagnostic and therapeutic issues [6]. For instance, teleconsultations seem to lead to greater frequency of contact between the physician and the patient, but appointments were shorter [7]. In addition, telemedicine reduced hospitalization rates when used for anticoagulation therapy consultations [8] or for malnutrition in older adults [9]. Nevertheless, the effectiveness of a telephone follow-up, particularly to detect severe cases of SARS-CoV-2 in Primary Health Care, has not been investigated in depth [10].

The objectives of this study were to analyze the effectiveness of telephone follow-up to detect severe cases of SARS-CoV-2 that progressed to pneumonia post-discharge and to identify complications of SARS-CoV-2 pneumonia in a 14-day follow-up.

## 2. Materials and Methods

### 2.1. Study Design and Population

The cohort study included individuals aged ≥15 years assigned to a primary healthcare center in Barcelona (Spain) with a total assigned population of 15,725 residents. Although the pandemic was not officially declared by the World Health Organization until 11 March 2020 [11], the sample for this study was recruited from 1 March (when the first case of SARS-CoV-2 was declared in our primary care setting) through 4 May 2020. All individuals who presented to the primary healthcare center with symptoms suggestive of SARS-CoV-2 were included in the follow-up protocol [5]. In addition, all patients with SARS-CoV-2 assigned to this primary healthcare center who went directly to the hospital were identified from the discharge registries and included in the cohort. Individuals were excluded who requested sick leave but did not present with suggestive symptoms or risk factors for SARS-CoV-2, who showed insufficient symptoms of SARS-CoV-2 according to medical criteria, or who had experienced symptoms more than 14 days before the study period. Moreover, those who reported close contact with an individual with SARS-CoV-2 more than 14 days before the study period or those who had been convalescing in hotels or nursing homes for more than 14 days were excluded.

### 2.2. SARS-CoV-2 Diagnosis and Follow-Up

The recommendation for all patients with symptoms suggestive of SARS-CoV-2 who presented to the primary healthcare center was home isolation, with written instructions provided for others in the household [12], along with treatment of symptoms and approval of sick leave as appropriate.

Telephone follow-up was performed on days 2, 4, 7, 10 and 14 after onset of SARS-CoV-2 symptoms; patients with risk factors for pneumonia received daily calls through day 5 and then the regularly scheduled calls. In each phone call, patients were asked about general symptomatology (fever, cough, dyspnea, asthenia, headache, myalgia, throat pain, nasal congestion, gastroenteritis, olfactory changes) and alarming symptoms for SARS-CoV-2 pneumonia (persistent cough, severe dyspnea, or fever >38.5 °C for more than 4 days) [5]. Patients were advised to call the primary healthcare center if any alarming symptom appeared between the scheduled follow-up calls. Whenever severe disease was suspected, the patient was referred for chest X-rays at a clinic or to the reference hospital, according to the medical criteria. Hospitalized patients were excluded from follow-up until discharge. A team of three general practitioners called all individuals diagnosed with pneumonia but non-hospitalized on days 2, 4, 7, 10 and 14 after diagnosis and hospitalized patients on days 1, 3, 7 and 14 after discharge. In each call, all patients were asked about the general symptomatology and specifically about the alarming symptoms. The patient was recalled the same day if he/she was not available at the scheduled time. Due to the high risk of pneumonia during the first week [5], if the call was unsuccessful the patient was called again twice each day until contact was made. Five patients with non-severe disease did not answer the phone calls; no hospital discharge registries or death certificates were found for this 1% loss to follow-up.

### 2.3. Variables Collected

Age, sex, SARS-CoV-2 risk factors: cardiopathy (ischemic heart disease, chronic arrhythmia, valvulopathy, myocardiopathy), type 2 diabetes mellitus, hypertension, history of immunosuppressive diseases or treatments (e.g., corticoids) and chronic obstructive pulmonary disease (COPD) were obtained from anonymized electronic medical records. In addition, the dates of symptoms onset, consultation at the primary healthcare center, chest X-ray, hospitalization and discharge were collected. Finally, the dates and number of calls per day, chest X-ray results and deaths during hospitalization were recorded.

### 2.4. Statistical Analysis

Continuous variables were summarized as mean (standard deviation) and categorical variables as proportions. Chi-square and Student’s *t*-tests were used as appropriate to compare the prevalence and means of different risk factors between patients with and without SARS-CoV-2 pneumonia, respectively. Differences in the incidence of pneumonia by specific risk factors, age, and other comorbidities at the end of follow-up were estimated with the log-rank test. The cumulative incidence function and unadjusted and age-adjusted hazard ratios were assessed by Cox regressions. Proportional hazards assumption was validated in all instances. All calculations were made with R statistical package (version 4.0.3; R Foundation for Statistical Computing, Vienna, Austria).

## 3. Results

The study included 453 patients, of which 430 (99%) completed the follow-up. The majority of patients with symptoms suggestive of SARS-CoV-2 made the first contact with the health system through the primary healthcare center (96%) and 82.8% of pneumonia diagnoses were done in this setting. More than 1 in 5 cases of non-severe disease (incidence rate = 20.8%) progressed to pneumonia (Figure 1). The cohort included slightly more women than men. Additionally, the prevalence of risk factors for severe SARS-CoV-2 was significantly higher in those with pneumonia (Table 1).

The mean elapsed time from symptoms onset to presentation at the primary healthcare center was 3 days. Patients were referred to hospital at a mean 7 days from symptom onset, with a mean hospital stay of 12 days (Figure 2). Of the 134 patients in the study sample with a chest X-ray performed in the primary care setting, 82 (61.2%) had pneumonia and 75 (56%) required hospitalization. Thus, 93.7% of individuals referred to hospital with pneumonia already had a radiology-based diagnosis.

We made 1798 telephone calls, 1498 to patients initially attended at the primary healthcare center and 300 in those diagnosed with pneumonia, with or without hospitalization. The mean number of calls in patients initially attended in the primary healthcare center was 3.4 (standard deviation 1.7). No hospital discharge registries were found for the five patients with non-severe disease who did not answer the phone calls. The diagnosis of pneumonia was most frequent on days 7 and 10 from symptoms onset. The highest effectiveness of telephone follow-up was observed between days 4 and 10, with 75.8% of pneumonias diagnosed in that period. The mean number of calls to patients after diagnosis of pneumonia (with or without hospitalization) was 3.5 (1.4). One patient presented with sudden death despite no record of complications or rehospitalization due to SARS-CoV-2 (Figure 3).

The crude cumulative incidence functions showed that individuals aged 65 years or older and patients with hypertension, COPD, or immunosuppression had significantly higher risk of SARS-CoV-2 pneumonia (Figure 4). In the multivariable analysis adjusted for age, the risk factor that remained significant was immunosuppression (hazard ratio (95% confidence interval) = 2.35 (1.08–5.09); *p*-value = 0.030) (Table 2).

## 4. Discussion

Telephone follow-up by primary healthcare professionals was effective and feasible to detect progression to severe SARS-CoV-2 disease and pneumonia. This procedure detected more than 80% of the pneumonias diagnosed. These patients received a chest X-ray and were rapidly referred to hospital with an appropriate radiology diagnosis. The short-term follow-up of individuals with pneumonia (with or without hospitalization) did not detect rehospitalizations or complications, except for one case of sudden death. Thus, the individuals hospitalized with pneumonia received close, long-term follow-up, with a mean hospital stay of 12 days, that decreased the efficiency of our 2-week telephone follow-up after discharge. The 2-week approach is appropriate for non-hospitalized individuals with pneumonia or those hospitalized who presented with severe complications (e.g., thrombotic disease). Nevertheless, our results must be interpreted within the context of the first wave of the pandemic, when no rapid diagnostic tests were yet available in primary care settings.

### 4.1. The Natural History of SARS-CoV-2

The disease evolution observed in our analysis concurs with the natural history of SARS-CoV-2 described in the literature [5]. A systematic review has confirmed a mean elapsed time of 7 days from symptoms onset until hospitalization due to pneumonia [13]. However, our work suggests another peak of pneumonia incidence on day 10 that pointed out the benefits of the schedule of phone calls (e.g., days 2, 4, 7, 10 and 14 in patients without risk factors).

Previous studies have shown that the most common risk factors in the development of SARS-CoV-2 infection are age ≥65 years, male sex, heart failure and cardiovascular diseases, hypertension, type 2 diabetes, chronic kidney disease, obesity and chronic obstructive pulmonary disease [14,15]. Moreover, a recent report has shown the high magnitude of the effect of immunosuppression [16], which was also significantly associated in our study, together with age. The remaining risk factors assessed did not have a significant association in our study because of the high influence of age and, in some cases, the lack of statistical power.

### 4.2. The Pandemic Accelerates the Digitalization of Healthcare

The ongoing expansion of telemedicine will accelerate technology-based solutions for tele-health, including remote monitoring of vital signs and acquisition of other health data in real time, thus enabling timely diagnosis and prompt initiation of treatment. Designation of new workflows enhanced by artificial intelligence will support an integrated transition between virtual and face-to-face care [17]. The large-scale demands of the pandemic caused a quick and substantial shift in how health care systems deliver care, forcing the incorporation of telemedicine into primary healthcare settings, with a huge increase in virtual visits and a decline in face-to-face attention [18]. Telemedicine will particularly reinforce in-home care, with great potential to ensure patient adherence to care plans, thus decreasing the risk of hospitalizations and associated costs and the potential for hospital-acquired infections and antimicrobial drug resistance. Effective in-home care for older adults can also translate into less disruption of social and family life, including the inconvenience of getting to appointments. In addition, the availability of telemedicine will encourage people to seek medical evaluation earlier in an illness, thus avoiding the detrimental consequences for the patient’s health and finances, as well as the health care system, of late diagnosis and treatment [17].

In the case of SARS-CoV-2, with the urgent need to avoid a dangerous situation such as hospital collapse, which was so plausible in this crisis, the telematic follow-up of patients with non-severe SARS-CoV-2 was assumed by primary healthcare settings [4]. This low-cost social-distancing strategy protects both healthcare professionals and patients and avoids overcrowding at the emergency room [19,20,21,22]. Despite the reduction in face-to-face medical visits, patients report a good level of satisfaction, even greater than the health professionals [23]. The opportunity to implement telemedicine that has been provided by the SARS-CoV-2 pandemic is unique, and several considerations are mandatory to avoid failures, including integration of information systems, a multidisciplinary approach, and legal issues [24,25]. Telematic follow-up includes several options: phone call, video call, web-based platforms, email, or mobile apps [26]. Telephone follow-up was implemented by the primary care system due to the universality and easy interaction allowed by this approach. Video-supported conversations (videochat) were also available if the patient consented to this option, but the abrupt outbreak of the pandemic did not permit proper development of other applications (apps) suitable for this purpose. Nevertheless, a rapid increase in the use of videoconference for healthcare purposes has been documented [27]. In addition, the use of smartphones to analyze breathing sounds could help with early diagnosis of the SARS-CoV-2 complications [28]. An on-going trial aims to evaluate the effectiveness of biosensors that register variables likely to predict disease prognosis [29]. Thus, adaptation to this new paradigm requires telematic tools scaled to population needs and useful to avoid the collapse of health services [19,30]. A recent scoping review points out that interactive environments represent an effective and costless tool for health services. These solutions offer real-time communication between clinics and patients, engaging them in ongoing dialogues beyond doctor consultations. This allows provision of healthcare information, follow-up of treatments and questions to be answered, and the development of more efficient and customer-oriented processes and of closer relationships with patients [31]. In addition, the digital platforms offer opportunity to present information about the participating organizations, to inform and keep the public up to date with themes of public interest, and to report about internal research activities [32].

Several small studies have already conducted telematic follow-up in patients with SARS-CoV-2 [33,34]. Although hospitalization was considered an end-point, those authors did not study the incidence of pneumonia, the time elapsed since hospitalization, or post-discharge progress. Lam et al. performed a telematic follow-up in a specific program including 50 patients, six (12%) of which were referred to hospital and four (8%) eventually hospitalized [33]. In another retrospective study with 48 patients that used a specific mobile app for the follow-up, six (12.5%) patients required hospitalization [34]. In these studies, the samples analyzed were not representative of the reference population, making it difficult to make any direct comparison with our results. Nevertheless, a review of evidence pointed out that approximately 14% of patients with SARS-CoV-2 require hospitalization [35]. In our cohort, this figure was increased to 20%.

### 4.3. Limitations

Our work had several limitations. On the one hand, the cases suggesting non-severe SARS-CoV-2 disease could not be confirmed because diagnostic tests were only available at hospitals at the time. Nevertheless, the epidemiological context and observation of the characteristic features of SARS-CoV-2 infection (e.g., anosmia), together with a low incidence of influenza during the study period (March–May), made the diagnosis of SARS-CoV-2 infection plausible in our cohort. The lack of test availability in primary care during the first wave meant that many cases with mild or no apparent symptoms were likely undetected. On the other hand, the use of discharge registries only identified patients who went directly to the hospital and were admitted. Thus, those with non-severe disease who did not require hospitalization were not initially included in our cohort. However, we were able to include in our analysis those who requested sick leave authorization, which requires approval by the assigned primary healthcare center. Finally, this study shows the effectiveness of active surveillance for SARS-CoV-2 in a primary care setting; however, implementation of this approach on a larger scale might be difficult. On the one hand, while telephone monitoring saves money compared to in-person contacts, the costs of any intensive patient supervision are high. On the other hand, additional discipline in follow-up may be required, both from health professionals and patients, to ensure the success of this model.

## 5. Conclusions

Telephone follow-up carried out from the primary healthcare setting was effective to detect pneumonia in individuals diagnosed with non-severe SARS-CoV-2. Our procedure ruled out those patients with non-severe disease and selected those who required hospital admission. Moreover, most of these patients arrived at the hospital with the needed radiology diagnosis already done. In analysis of anonymized data, the factors significantly associated with disease progression were an age ≥65 years and the presence of immunosuppression. To avoid the collapse of the health system, the digitalization accelerated by the present pandemic must be consolidated. Nevertheless, measures that have shown high efficacy to reduce propagation, such as physical distancing, face masks and eye protection, will be required until vaccination can achieve so-called herd immunity. The rapid increase and wide adaptation of telemedicine into care delivery models should be balanced against the perception by some patients and providers of uncertain safety and value compared with face-to-face care. Effective and efficient virtual care will require a reliable communication infrastructure and affordable, readily accessible broadband connectivity to all regions.

## Figures and Tables

**Figure 1 healthcare-09-01548-f001:**
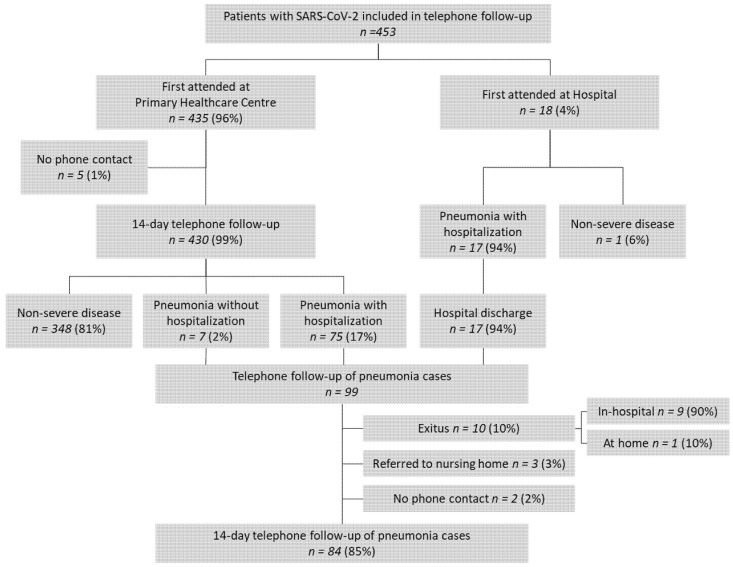
Flow chart of the study.

**Figure 2 healthcare-09-01548-f002:**
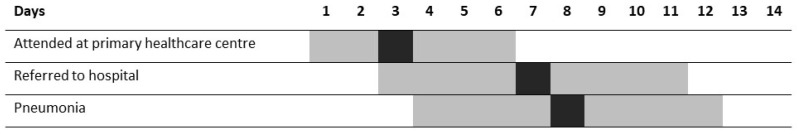
Days elapsed until consultation at primary healthcare center, hospital referral and diagnosis of pneumonia. Mean (dark gray) and standard deviation (light gray).

**Figure 3 healthcare-09-01548-f003:**
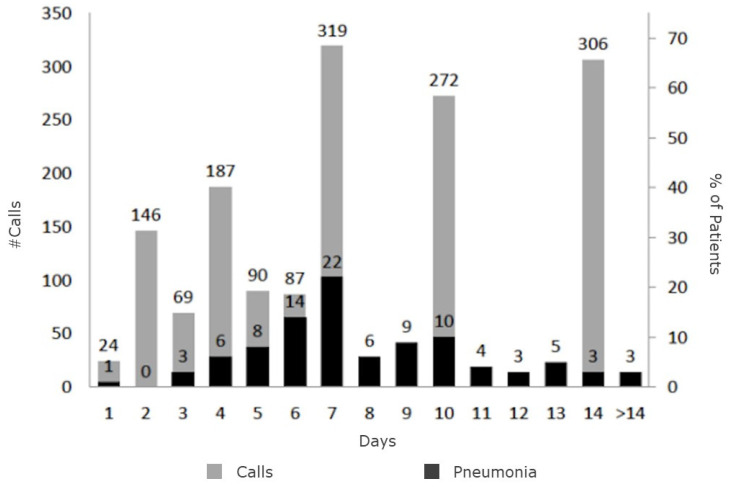
Calls performed and development of SARS-CoV-2 pneumonia in individuals with non-severe disease.

**Figure 4 healthcare-09-01548-f004:**
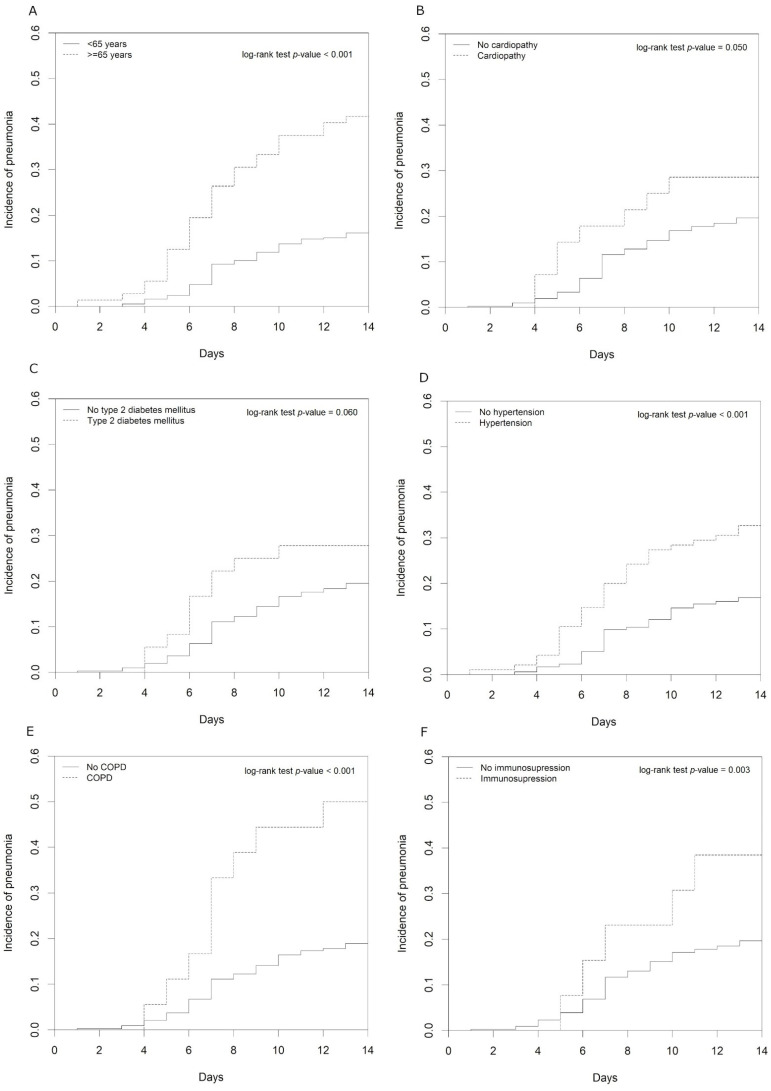
Cumulative incidence function for SARS-CoV-2 pneumonia by age (<65 and ≥65 years) (**A**), cardiopathy (**B**), type 2 diabetes mellitus (**C**), hypertension (**D**), chronic obstructive pulmonary disease (COPD) (**E**) and immunosuppression (**F**).

**Table 1 healthcare-09-01548-t001:** Characteristics of patients with SARS-CoV-2 infection included in the cohort.

	All (*n* = 453)	SARS-CoV-2 Non-Severe Disease (*n* = 354)	SARS-CoV-2 Pneumonia (*n* = 99)	*p*-Value
Age, mean (SD)	50 (16)	47 (15)	60 (15)	<0.001
Sex (male), *n* (%)	189 (41.7)	143 (40.4)	46 (46.5)	0.333
SARS-CoV-2 risk factors				
Age ≥65 years	74 (16.3)	37 (10.5)	37 (37.4)	<0.001
Cardiopathy	30 (6.6)	17 (4.8)	13 (13.1)	0.007
Type 2 diabetes mellitus	37 (8.2)	22 (6.2)	15 (15.2)	0.008
Hypertension	97 (21.4)	58 (16.4)	39 (39.4)	<0.001
Immunosuppression	18 (4.0)	9 (2.5)	9 (9.1)	0.007
Chronic obstructive pulmonary disease	13 (2.9)	6 (1.7)	7 (7.1)	0.010

**Table 2 healthcare-09-01548-t002:** Cox regression models for the incidence of SARS-CoV-2 pneumonia.

	Unadjusted HR (95% CI)	*p*-Value	Adjusted HR * (95% CI)	*p*-Value
Age ≥ 65 years	3.31 (2.16; 5.08)	<0.001	--	--
Cardiopathy	1.90 (0.99; 3.65)	0.055	0.93 (0.48; 1.82)	0.841
Type 2 diabetes mellitus	1.79 (0.98; 3.28)	0.058	0.94 (0.50; 1.77)	0.858
Hypertension	2.40 (1.58; 3.66)	<0.001	1.21 (0.74; 1.98)	0.446
Chronic obstructive pulmonary	3.20 (1.61; 6.37)	0.001	1.23 (0.57; 2.62)	0.597
Immunosuppression	3.03 (1.40; 6.54)	0.005	2.35 (1.08; 5.09)	0.030

* Model adjusted for age. CI: Confidence interval. HR: Hazard Ratio.

## Data Availability

The data presented in this study are available on request from the corresponding author.

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
