# Peer review of "Effectiveness of Telephone Monitoring in Primary Care to Detect Pneumonia and Associated Risk Factors in Patients with SARS-CoV-2"

_healthcare, 2021, doi:10.3390/healthcare9111548_

Round 1

Reviewer 1 Report

Thank you for inviting me to revise this paper titled Effectiveness of phone monitoring in Primary Care to detect 2 pneumonia in patients with SARS-CoV-2.

This study addresses an interesting topic, i.e. telemedicine and specifically it investigates the use of a technological tool (i.e. phone calls) to monitor the progression of infection in individuals with the Sars-CoV-2.

Nevertheless the paper needs to be completely revised to be published in a scientific journal. First of all, the authors should demonstrate a knowledge of the existing literature on telemedicine and use of digital tools in the health industry (a comprehensive review is provided by Pianese T, Belfiore P. Exploring the Social Networks’ Use in the Health-Care Industry: A Multi-Level Analysis. International Journal of Environmental Research and Public Health. 2021; 18(14):7295. https://doi.org/10.3390/ijerph18147295).

This would enable the author a) to better position their study; b) evidence its originality and innovativeness; c) discuss their findings according to existing researches and thus point out the contribution to the advancement of the existing knowledge on the topic of telemedicine.

Author Response

Reviewer 1

Thank you for inviting me to revise this paper titled Effectiveness of phone monitoring in Primary Care to detect 2 pneumonia in patients with SARS-CoV-2.

This study addresses an interesting topic, i.e. telemedicine and specifically it investigates the use of a technological tool (i.e. phone calls) to monitor the progression of infection in individuals with the Sars-CoV-2.

Nevertheless the paper needs to be completely revised to be published in a scientific journal. First of all, the authors should demonstrate a knowledge of the existing literature on telemedicine and use of digital tools in the health industry (a comprehensive review is provided by Pianese T, Belfiore P. Exploring the Social Networks’ Use in the Health-Care Industry: A Multi-Level Analysis. International Journal of Environmental Research and Public Health. 2021; 18(14):7295. https://doi.org/10.3390/ijerph18147295).

This would enable the author a) to better position their study; b) evidence its originality and innovativeness; c) discuss their findings according to existing researches and thus point out the contribution to the advancement of the existing knowledge on the topic of telemedicine.

Reply: Thank you. We have added a new paragraph with two new references, including the scoping review noted by the reviewer, in Discussion (page 7, line 213-220): “A recent scoping review points out that interactive environments represent an effective and costless tool for health services. These solutions offer real-time communication between clinics and patients, engaging them in ongoing dialogues beyond doctor consultations. This allows provision of healthcare information, follow-up of treatments and questions to be answered, and the development of more efficient and customer-oriented processes and of closer relationships with patients (Pianese T. Int J Environ Res Public Health 2021). In addition, the digital platforms offer opportunity to present information about the participating organizations, to inform and keep the public up to date with themes of public interest, and to report about internal research activities (Thackeray R. BMC Public Health. 2012)”.

We have added the following new references (#26 and #27):

Pianese, T., Belfiore, P. Exploring the Social Networks' Use in the Health-Care Industry: A Multi-Level Analysis. Int J Environ Res Public Health 2021, 18(14), 7295.

Thackeray, R., Neiger, B. L., Smith, A. K., Van Wagenen, S. B. Adoption and use of social media among public health departments. BMC Public Health 2012, 12, 242.

Reviewer 2 Report

Firstly, I thank Editorial Committee for the opportunity to review this manuscript. The authors deal with an important issue during the COVID-19 pandemic related to pneumonia detection in patients. Furthermore, the proposed manuscript meets adequately the purposes of the journal. The methodological rigour is adequate, using the appropriate statistical tests, recruiting a large sample, and performing several follow-up periods. Moreover, the figures and tables shown are suitable, especially Figure 2 and 4. However, below I suggest some recommendations and suggestions to improve the quality of their manuscript:

  • The study began on March 1, 2020, before the official declaration of the pandemic by the WHO and the population confinement in Spain. The authors should clarify why they began to collect data before these events.
  • The use of video consultations has been rapidly developed during the pandemic. Please include this issue in 4.2. section (p. 7, line 183 – 216). I suggest you the following reference: https://pubmed.ncbi.nlm.nih.gov/32679848/

Jiménez-Rodríguez D, Santillán García A, Montoro Robles J, Rodríguez Salvador MDM, Muñoz Ronda FJ, Arrogante O. Increase in Video Consultations During the COVID-19 Pandemic: Healthcare Professionals' Perceptions about Their Implementation and Adequate Management. Int J Environ Res Public Health. 2020;17(14):5112. Published 2020 Jul 15. doi:10.3390/ijerph17145112.

  • According to the Guide for Authors of the journal, references should be listed by Arabic numerals, not by roman numerals. Please, correct this issue throughout the manuscript and in the reference list.
  • Lastly, all references are updated corresponding to the last two years.

Author Response

Reviewer 2

Firstly, I thank Editorial Committee for the opportunity to review this manuscript. The authors deal with an important issue during the COVID-19 pandemic related to pneumonia detection in patients. Furthermore, the proposed manuscript meets adequately the purposes of the journal. The methodological rigour is adequate, using the appropriate statistical tests, recruiting a large sample, and performing several follow-up periods. Moreover, the figures and tables shown are suitable, especially Figure 2 and 4. However, below I suggest some recommendations and suggestions to improve the quality of their manuscript:

The study began on March 1, 2020, before the official declaration of the pandemic by the WHO and the population confinement in Spain. The authors should clarify why they began to collect data before these events.

Reply: Although the official declaration of pandemic was dated as March 11, 2020, the first case of SARS-CoV-2 in the city of Barcelona was declared on February 26 (Interactive Map of COVID-19 Cases [Mapa interactiu de casos per ABS] available at http://aquas.gencat.cat/ca/actualitat/ultimes-dades-coronavirus/mapa-per-abs/). We began recruitment with the first case detected in our specific primary care setting (March 1st). We appreciate the opportunity to clarify this point in the new version of the manuscript (page 2, lines 60-64): “Although the pandemic was not officially declared by the World Health Organization until March 11, 2020 (World Health Organization. 2020), the sample for this study was recruited from March 1 (when the first case of SARS-CoV-2 was declared in our primary care setting) through May 4, 2020. All individuals who presented to the primary healthcare center with symptoms suggestive of SARS-CoV-2 were included in the follow-up protocol (Struyf T. Cochrane Database Syst Rev. 2020)”.

We have added a new reference (#7):

World Health Organization. WHO announces COVID-19 outbreak a pandemic. Available at:  https://www.euro.who.int/en/health-topics/health-emergencies/coronavirus-covid-19/news/news/2020/3/who-announces-covid-19-outbreak-a-pandemic (26 October 2021, date last accessed).

The use of video consultations has been rapidly developed during the pandemic. Please include this issue in 4.2. section (p. 7, line 183 – 216). I suggest you the following reference: https://pubmed.ncbi.nlm.nih.gov/32679848/

Jiménez-Rodríguez D, Santillán García A, Montoro Robles J, Rodríguez Salvador MDM, Muñoz Ronda FJ, Arrogante O. Increase in Video Consultations During the COVID-19 Pandemic: Healthcare Professionals' Perceptions about Their Implementation and Adequate Management. Int J Environ Res Public Health. 2020;17(14):5112. Published 2020 Jul 15. doi:10.3390/ijerph17145112.

Reply: Thank you. We have added the suggested reference (#22) on page 7, lines 204-209: “Telephone follow-up was implemented by the primary care system due to the universality and easy interaction allowed by this approach. Video-supported conversations (videochat) were also available if the patient consented to this option, but the abrupt outbreak of the pandemic did not permit proper development of other applications (apps) suitable for this purpose. Nevertheless, a rapid increase in the use of videoconference for healthcare purposes has been documented (Jimenez-Rodriguez D. Int J Environ Res Public Health. 2020)”.

According to the Guide for Authors of the journal, references should be listed by Arabic numerals, not by roman numerals. Please, correct this issue throughout the manuscript and in the reference list.

Reply: Thank you, we have corrected this error.

Lastly, all references are updated corresponding to the last two years.

Reply: Thank you.

Reviewer 3 Report

General comments

  1. Does cohort study need the approval of the research ethics committee and/or is the patient consent needed?
  2. The telephone follow-up in the case of hospitalized patients is not relevant, those cases should be excluded from the study or discussed separately.
  3. The title of the paper is not in concordance with the content of the paper, because a significant part of the manuscript is about the correlation between the severity of the COVID-disease and different comorbidities; and in this context the presence or absence of the telematic follow-up in patients is not relevant.

Other comments:

  1. Row 123: where did the value of 97.3% came from?
  2. Row 109-124 and Figure 1: if a total number of 99 patients were diagnosed with pneumonia, but only 80 cases were diagnosed by X-chest ray, in the other cases how was the diagnosis set?
  3. Table 1: In the bracket the percentages are presented?
  4. Figure 3: The ox and oy axis titles are missing
  5. Row 186: “settings.4” should be replaced with “settings.“.
  6. Row 192: formatting of “[xvii.” should be revised.
  7. Row 211: “hospitalized.24” should be replaced “hospitalized.”
  8. Row 212: “hospitalization.25” should be replaced with “hospitalization.”
  9. Row 216: formatting of “[xxvi.” should be revised

Author Response

Reviewer 3

General comments

Does cohort study need the approval of the research ethics committee and/or is the patient consent needed?

Reply: The study protocol was approved by the local ethics committee (Fundació IDIAP Jordi Gol, #20/087-PCV). A clinical protocol was implemented for “COVID-safe” telephone follow-up of our primary care patients presenting with symptoms. When the follow-up was completed, we collected the study variables for analysis. Informed consent was not required because the manuscript was based on anonymized review of electronic medical records. We have added this clarification to the IRB section.

The telephone follow-up in the case of hospitalized patients is not relevant, those cases should be excluded from the study or discussed separately.

Reply: Individuals hospitalized due to pneumonia were excluded from the initial follow-up but this subsample was included in post-discharge follow-up. This point has been better explained in the new version of the manuscript for the sake of clarity (page 2, line 86-87): “Hospitalized patients were excluded from follow-up until discharge”.

The title of the paper is not in concordance with the content of the paper, because a significant part of the manuscript is about the correlation between the severity of the COVID-disease and different comorbidities; and in this context the presence or absence of the telematic follow-up in patients is not relevant.

Reply: Thank you. The title of the new version of the manuscript is: “Effectiveness of telephone monitoring in Primary Care to detect pneumonia and associated risk factors in patients with SARS-CoV-2”

Other comments:

Row 123: where did the value of 97.3% came from?

Reply: Thank you. We have corrected the typo; the correct value is 93.7%. This figure is the percentage of individuals who required hospitalization for pneumonia among all those with who had a diagnostic chest X-ray performed in a primary care setting before their arrival to hospital [i.e. (75 / 80) x 100].

Row 109-124 and Figure 1: if a total number of 99 patients were diagnosed with pneumonia, but only 80 cases were diagnosed by X-chest ray, in the other cases how was the diagnosis set?

Reply: We performed 80 chest X-rays in the primary care setting; thus, most patients arrived to hospital with this test already done. The tests performed to confirm the diagnosis of individuals first attended at hospital were not collected in our study, although this was most likely a chest X-ray. We have rewritten the paragraph for the sake of clarity (page 3, lines 126-128): “Of the 134 patients in the study sample with a chest X-ray performed in the primary care setting, 82 (61.2%) had pneumonia and 75 (56%) required hospitalization”.

Table 1: In the bracket the percentages are presented?

Reply: Thank you. We have homogenized the presentation of percentages in Figure 1 and added a new “Hospital discharge” box

Figure 1. Flow chart of the study

Figure 3: The ox and oy axis titles are missing

Reply: We apologize for this oversight and have included the axis name in the new version of Figure 3. We also appreciate the reviewer’s attention to typos in the tables and have corrected them as noted.

Figure 3. Calls performed and development of SARS-CoV-2 pneumonia in individuals with non-severe disease.

Row 186: “settings.4” should be replaced with “settings.“.

Reply: We have corrected the typo.

Row 192: formatting of “[xvii.” should be revised.

Reply: We have corrected the typo.

Row 211: “hospitalized.24” should be replaced “hospitalized.”

Reply: We have corrected the typo.

Row 212: “hospitalization.25” should be replaced with “hospitalization.”

Reply: We have corrected the typo.

Row 216: formatting of “[xxvi.” should be revised

Reply: We have corrected the typo.

Reviewer 4 Report

Article: Effectiveness of phone monitoring in Primary Care to detect pneumonia in patients with SARS-CoV-2

I read the article with great curiosity. Well-described methodology. The study represents a type of epidemiological surveillance that we call active surveillance, and by definition it is very effective. It is characterized by high sensitivity, unlike passive supervision. Certainly, the study may indicate the advantages of such supervision, while its implementation is probably impossible on a larger scale (e.g. on a national scale). The costs of this type of supervision are always high. You have to remember about patients who did not answer the phone, and keep trying to contact you;

On the other hand, telephone control may be cheaper than face-to-face contacts, but it probably requires more discipline, both from medical staff and patients.

I think that this aspect can be added to the section by the authors of the study's limitations, or for discussion (maybe this is not a limitation of the study in itself, but a limitation regarding the possibility of implementing such a procedure on a larger scale); I leave for discussion;

Thank you,

Author Response

Reviewer 4

I read the article with great curiosity. Well-described methodology. The study represents a type of epidemiological surveillance that we call active surveillance, and by definition it is very effective. It is characterized by high sensitivity, unlike passive supervision. Certainly, the study may indicate the advantages of such supervision, while its implementation is probably impossible on a larger scale (e.g. on a national scale). The costs of this type of supervision are always high. You have to remember about patients who did not answer the phone, and keep trying to contact you;

On the other hand, telephone control may be cheaper than face-to-face contacts, but it probably requires more discipline, both from medical staff and patients.

I think that this aspect can be added to the section by the authors of the study's limitations, or for discussion (maybe this is not a limitation of the study in itself, but a limitation regarding the possibility of implementing such a procedure on a larger scale); I leave for discussion;

Reply: Thank you. We have included this point in Limitations (page 8, lines 257-263): “Finally, this study shows the effectiveness of active surveillance for SARS-CoV-2 in a primary care setting; however, implementation of this approach on a larger scale might be difficult. On the one hand, while telephone monitoring save money compared to in-person contacts, the costs of any intensive patient supervision are high. On the other hand, additional discipline in follow-up may be required, both from health professionals and patients, to ensure the success of this model”.

Round 2

Reviewer 1 Report

Despite authors addressed a number of concerns, the study still require some improvements.

Authors should revise the abstract by reporting the objective of the study, methodology and findings of the study. It could be important also to highlight the innovativeness and originality of the study. In this revised draft the authors report in the abstract how many people were contacted, how many people were hospitalized etc. In my opinion, the objective of the study should not be narrow as it has been presented in the actual draft. Rather it should point out at the role of Information and Communication Technology in the health industry.

Put more effort in the discussion section to stress how your study is contributing to the advancement of the knowledge on this theme.

Author Response

Reviewer 1

Despite authors addressed a number of concerns, the study still requires some improvements.

Authors should revise the abstract by reporting the objective of the study, methodology and findings of the study. It could be important also to highlight the innovativeness and originality of the study. In this revised draft the authors report in the abstract how many people were contacted, how many people were hospitalized etc. In my opinion, the objective of the study should not be narrow as it has been presented in the actual draft. Rather it should point out at the role of Information and Communication Technology in the health industry.

Put more effort in the discussion section to stress how your study is contributing to the advancement of the knowledge on this theme.

Reply: Thank you. We have modified several parts of the manuscript to point out the role of Information and Communication Technology in health care.

Abstract:

Improved technology facilitates the acceptance of telemedicine. The aim was to analyze the effectiveness of telephone follow-up to detect severe SARS-CoV-2 cases that progressed to pneumonia. A prospective cohort study with 2-week telephone follow-up was carried out March 1 to May 4, 2020, in a primary healthcare center in Barcelona. Individuals aged ≥15 years with symptoms of SARS-CoV-2 were included. Outpatients with non-severe disease were called on days 2, 4, 7, 10 and 14 after diagnosis; patients with risk factors for pneumonia received daily calls through day 5 and then the regularly scheduled calls. Patients hospitalized due to pneumonia received calls on days 1, 3, 7 and 14 post-discharge. Of the 453 included patients, 435 (96%) were first attended at a primary healthcare center. The 14-day follow-up was completed in 430 patients (99%), with 1798 calls performed. Of the 99 cases of pneumonia detected (incidence rate 20.8%), one third appeared 7 to 10 days after onset of SARS-CoV-2 symptoms. Ten deaths due to pneumonia were recorded. Telephone follow-up by a primary healthcare center was effective to detect SARS-CoV-2 pneumonias and to monitor related complications. Thus, telephone appointments between a patient and their health care practitioner benefit both health outcomes and convenience.

Introduction (page 2, lines 49-58):

To strengthen lockdown precautions and avoid the exposure of other patients and healthcare professionals, a telemedicine strategy (i.e., telephone follow-up) was prioritized. A growing body of evidence supports the safety and efficacy of telemedicine, showing equivalencies to a conventional medical appointment both in diagnostic and therapeutic issues (Shigekawa E. 2018). For instance, teleconsultations seem to lead to greater frequency of contact between the physician and the patient, but appointments were shorter (Downes MJ. 2017). In addition, telemedicine reduced hospitalization rates when used for anticoagulation therapy consultations (Lee M. 2018) or for malnutrition in older adults (Marx W. 2018). Nevertheless, the effectiveness of a telephone follow-up, particularly to detect severe cases of SARS-CoV-2 in Primary Health Care, has not been investigated in depth (Gao Y).

New references added:

Shigekawa, E., Fix, M., Corbett, G., Roby, D. H., Coffman, J. The Current State Of Telehealth Evidence: A Rapid Review. Health Aff (Millwood) 2018, 37(12), 1975–1982.

Downes, M. J., Mervin, M. C., Byrnes, J. M., Scuffham, P. A. Telephone consultations for general practice: a systematic review. Syst Rev 2017, 6(1), 128.

Lee, M., Wang, M., Liu, J., Holbrook, A. Do telehealth interventions improve oral anticoagulation management? A systematic review and meta-analysis. J Thromb Thrombolysis 2018, 45(3), 325–336.

Marx, W., Kelly, J. T., Crichton, M., Craven, D., Collins, J., Mackay, H., Isenring, E., Marshall, S. Is telehealth effective in managing malnutrition in community-dwelling older adults? A systematic review and meta-analysis. Maturitas 2018, 111, 31–46.

Discussion (page 6, lines 197-217)

The ongoing expansion of telemedicine will accelerate technology-based solutions for tele-health, including remote monitoring of vital signs and acquisition of other health data in real time, thus enabling timely diagnosis and prompt initiation of treatment. Designation of new workflows enhanced by artificial intelligence will support an integrated transition between virtual and face-to-face care (Temesgen, Z. M. 2020). The large-scale demands of the pandemic caused a quick and substantial shift in how health care systems deliver care, forcing the incorporation of telemedicine into primary healthcare settings, with a huge increase in virtual visits and a decline in face-to-face attention (Joy, M. 2020). Telemedicine will particularly reinforce in-home care, with great potential to ensure patient adherence to care plans, thus decreasing the risk of hospitalizations and associated costs and the potential for hospital-acquired infections and antimicrobial drug resistance. Effective in-home care for older adults can also translate into less disruption of social and family life, including the inconvenience of getting to appointments. In addition, the availability of telemedicine will encourage people to seek medical evaluation earlier in an illness, thus avoiding the detrimental consequences for the patient’s health and finances, as well as the health care System, of late diagnosis and treatment (Temesgen, Z. M. 2020).

In the case of SARS-CoV-2, with the urgent need to avoid a dangerous situation such as hospital collapse that was so plausible in this crisis, the telematic follow-up of patients with non-severe SARS-CoV-2 was assumed by primary healthcare settings (Krist, A. H. 2020). This low-cost social-distancing strategy protects both healthcare professionals and patients and avoids overcrowding at the emergency room (Pérez Sust, P. 2020; Vidal-Alaball, J. Aten Primaria 2020; Khairat, S. 2020; Chu, D. K. Lancet 2020).

New reference added:

Temesgen, Z. M., DeSimone, D. C., Mahmood, M., Libertin, C. R., Varatharaj Palraj, B. R., Berbari, E. F. Health Care After the COVID-19 Pandemic and the Influence of Telemedicine. Mayo Clin Proc 2020, 95(9S), S66–S68.

Conclusion (page 9, lines 294-299)

The rapid increase and wide adaptation of telemedicine into care delivery models should be balanced against the perception by some patients and providers of uncertain safety and value compared with face-to-face care. Effective and efficient virtual care will require a reliable communication infrastructure and affordable, readily accessible broadband connectivity to all regions.
